# Innovative Combinations, Cellular Therapies and Bispecific Antibodies for Chronic Lymphocytic Leukemia: A Narrative Review

**DOI:** 10.3390/cancers16071290

**Published:** 2024-03-26

**Authors:** Andrea Visentin, Sara Frazzetto, Livio Trentin, Annalisa Chiarenza

**Affiliations:** 1Hematology Unit, Department of Medicine, University of Padova, 35128 Padova, Italy; andrea.visentin@unipd.it; 2Hematology and Stem Cell Transplantation Unit, A.O.U. Policlinico, 95123 Catania, Italy; sara.frazzetto@gmail.com (S.F.); annalisa.chiarenza@gmail.com (A.C.)

**Keywords:** CLL, BTK, CAR-T, BsAb, BCL2

## Abstract

**Simple Summary:**

Chronic Lymphocytic Leukemia (CLL) is one of the most frequent leukemia in the west countries for adult patients. It’s typically a slow-growing disease and the majority of patients do not need immediate treatment at diagnosis. Advanced stages or symptomatic disease require therapy. The treatment landscape of CLL has changed considerably in the last decades with the introduction of new targeted agents leading to improved outcome for patients with CLL compared to standard chemo-immunotherapy, especially for those with high-risk features, as del17p13, TP53 mutations and unmutated immunoglobulin heavy chain (IGHV) genes. In this narrative review, we comprehensively summarized and discussed all the new approaches currently investigated in completed and on-going clinical trials, both with single new agents and in combination strategies to pursuing not only a disease control but the eradication of the leukemic clone.

**Abstract:**

In the last few years, several agents targeting molecules that sustain the survival and the proliferation of chronic lymphocytic leukemia (CLL) cells have become clinically available. Most of these drugs target surface proteins, such as CD19 or CD20, via monoclonal or bispecific monoclonal antibodies (BsAbs), CAR T cells, intracellular proteins like BTK by using covalent or non-covalent inhibitors or BCL2 with first or second generation BH3-mimetics. Since the management of CLL is evolving quickly, in this review we highlighted the most important innovative treatments including novel double and triple combination therapies, CAR T cells and BsAbs for CLL. Recently, a large number of studies on novel combinations and newer strategic options for CLL therapy have been published or presented at international conferences, which were summarized and linked together. Although the management of treatment with a single continuous agent is easier, the emergence of protein mutations, long-term toxicities and costs are important concerns that favor the use of a fixed duration therapy. In the future, a measurable residual disease (MRD)-guided treatment cessation and MRD-based re-initiation of targeted therapy seems to be a more feasible approach, allowing identification of the patients who might benefit from continuous therapy or who might need a consolidation with BsAbs or CAR T cells to clear the neoplastic clone.

## 1. Introduction

The treatment landscape of chronic lymphocytic leukemia (CLL) has been significantly changed in the last 10 years, moving from chemoimmunotherapy for all patients, to chemoimmunotherapy for only young and fit patients without adverse prognostic markers, to continuous single-target drugs and, lastly, to fixed-duration (FD) combination therapy.

Given the pivotal role of the B-cell receptor (BCR) in sustaining the survival and the proliferation of CLL cells, most targeted agents have been developed against Bruton tyrosine kinase (BTK) and BCL2 (B-cell lymphoma 2) (Figure 1). Both class of drug represent the cornerstones of the contemporary paradigm shift of treatment in CLL and other B-cell lymphoproliferative malignancies.

Several studies have confirmed a benefit for the entire CLL population in terms of progression-free survival (PFS) and in some cases also of overall survival (OS) compared to chemoimmunotherapy [1,2,3,4,5,6]. Even if, in many studies, the weakness of the control arm, such as chlorambucil or ofatumumab as single agents, has generated several uncertainties [7,8], the advantage of targeted drugs remains undisputed to the point that the chemo-free approach has supplanted traditional chemoimmunotherapy. While today very a few patients are treated with chemoimmunotherapy (CIT), given the higher efficacy and better safety profile of targeted drugs [9], one of the most unresolved question is which patients might be candidates for continuous treatment rather than a FD therapy, as the former is associated with several concerns regarding adverse events, long-term safety, emergence of resistance due to protein mutations and, last but not the least, financial toxicities, in particular for public health care systems.

**Figure 1 cancers-16-01290-f001:**
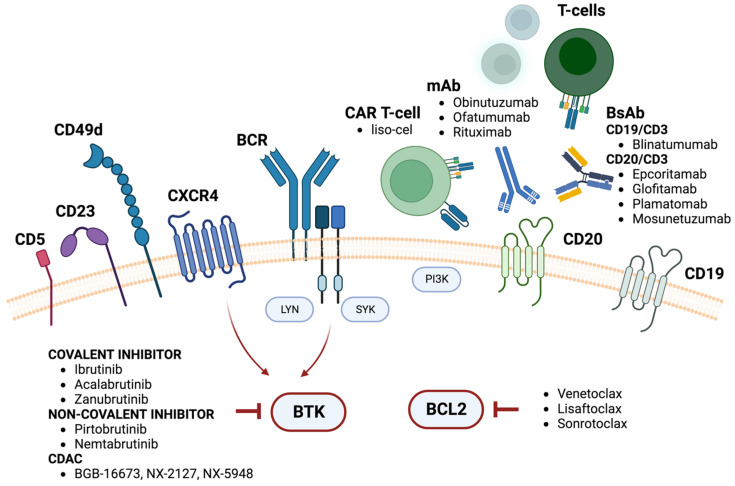
Current and novel targeted therapies in CLL. The neoplastic cells of chronic lymphocytic leukemia express on their surfaces CD5, CD20 (even if a low levels), CD23 and CD200. CD20 can be targeted by anti-CD20 monoclonal antibodies (mAbs) like rituximab, ofatumumab or obinutuzumab. These cells move from the peripheral blood to the bone morrow and secondary lymphoid organs thanks to integrins, such CD49d, cytokines and chemokine receptors, like CXCR4 [10]. One of the main membrane receptors is the B-cell receptor (BCR), which after antigen engagement actives several kinases such as Lyn, Syk, PI3K and BTK, which play a key role in mediating the survival and the proliferation of CLL cells [11,12]. BTK (Bruton’s Tyrosine kinase) can be targeted by covalent inhibitors such as ibrutinib, acalabrutinib and zanubrutinib but also the non-covalent inhibitors pirtobrutinib and nemtabrutinib and CDACs (chimeric degradation activation compounds) like BGB-16673, NX-2127 and NX-5948 [13,14]. CLL cells are also characterized by the overexpression of the anti-apoptotic protein BCL2, mainly due to the loss of mir15-a and mir16-1, whose main target is the mRNA derived from the *BCL2* gene. The BH3 mimetics, such as venetoclax, sonrotoclax and lisaftoclax are able to disrupt the binding of BCL2 with BID, triggering apoptosis. The tumor microenvironment represented by the nurse-like cells, macrophages, mesenchymal stromal cells and T-cells is another important player in sustaining the survival of CLL cells and favoring the resistance to therapies. Bispecific antibodies (BsAbs), such as blinatumomab, mosunetuzumab, glofitamab epcoritamab and plamatomab favor the activation of T cells against neoplastic cells. In a similar way, chimeric antigen receptors (CARs), once expressed by T cells (CAR T-cells), such as lisocabtagene maraleucel (liso-cel), improve the T-cell killing of CLL cells.

FD therapy has been explored in previously untreated patients with CLL by using two different strategies, combing venetoclax with either an anti-CD20 monoclonal antibody (Ab, i.e., rituximab or obinutuzumab) or a BTK inhibitor (BTKi, such as ibrutinib, acalabrutinib or zanubrutinib). In addition, in some trials the treatment was discontinued after a pre-defined time, usually 12 or 15 months, whereas others were designed with a measurable residual disease (MRD)-driven discontinuation. Furthermore, there are also uncertainties about triple combinations, even if some results are very promising.

Although most patients relapsing after FD therapy did not show BTK or BCL2 mutations, outside of clinical trials the number of patients failing with both a BTKi and a BCL2i are increasing and the outcome of these double-exposed patients is still poor [15,16]. Encouraging results are also coming with the use of cellular therapies and bispecific antibodies (BsAbs).

In this review, we summarized the most important novel combinations, cellular therapies and bispecific antibodies in CLL.

## 2. Fixed-Duration Double Therapies in Treatment-Naive Patients

In the VERITAS, CLL13, HOVON 139/GiVe, MDACC, CAPTIVATE, FLAIR and BGB-11417-101 trials, FD treatment combinations were assessed in young CLL patients (Table 1). Conversely, the SEQUOIA, GLOW and CLL14 trials dealt with elderly patients (Table 1).

The Italian GIMEMA (Gruppo Italiano per lo studio delle Malattie Ematologiche dell’Adulto) phase 2 LLC1518 VERITAS trial investigated the efficacy and safety of a 12-month FD of venetoclax and rituximab (VenR) in young (≤65 years), fit patients with CLL and high-risk features such as unmutated IGHV (U-IGHV) and/or TP53 abnormalities (abn, deletion and/or mutation) [17]. Seventy-five patients were enrolled, the median age was 54 years, ranging from 38 to 65 years, 96% had unmutated IGHV and 12% had TP53 abn. The overall response rate was 95%, with a complete remission (CR) rate of 76%. Allele-specific PCR assay MRD was undetectable (uMRD4, i.e., <10^−4^) in the peripheral blood (PB) of 69% of patients and in the bone marrow (BM) of 59% of patients. The study was updated at the 2023 ASH meeting, showing that after 33 months, uMRD4 was maintained by almost half of the patients who achieved it [18]. In multivariate analysis, the factors associated with a significantly shorter MRD-free survival were the presence of U-IGHV, TP53 abn or advanced stage, while the clinical response did not have a significant impact. The 36-month OS was 96% [18]. Almost half of the patients had at least grade 3 or higher adverse events, mainly neutropenia 37% and infections 12%.

In the phase 3 GAIA-CLL13 trial, fit CLL patients without TP53 abn were randomly assigned to receive 6 cycles of chemoimmunotherapy (CIT, fludarabine–cyclophosphamide–rituximab (FCR) or bendamustine–rituximab (BR)) or 12 cycles of VenR, VG (venetoclax and obinutuzumab) or VG–ibrutinib (GIV) [19]. Overall, 926 patients with a cumulative illness rating scale (CIRS) ≤ 6, a normal creatinine clearance and a ECOG performance status of 0–2 were assigned to one of the four treatments and their characteristics were well balanced among arms. At month 15, the percentage of patients with uMRD4 by next-generation sequencing (NGS) in the PB and BM was significantly higher in the obinutuzumab, VG (87% and 73%, *p* < 0.001) and VIG arms (92% and 78%, *p* < 0.001) than in the VenR (57% and 43%) or chemoimmunotherapy arms (52% and 37%). Furthermore, the majority of patients treated with GV or GIV (60% and 66%) achieved uMRD6. After a median follow-up of 39 months, the 3-year PFS was 91% with the triple therapy, 88% with VG, 81% with VenR and 76% in the FCR/BR arm (*p* < 0.001). Both IGHV unmutated and mutated patients relapsed early if treated with VenR, suggesting that the anti-CD20 mAb type is important to achieve deep and sustained responses. The updated results presented at the 2023 ASH congress showed that U-IGHV patients and those with a bulky disease had a shorter PFS in the pooled GV/GIV group [20]. In addition, highly complex karyotypes (≥5 chromosomal aberrations) and translocations are independent prognostic factors for inferior PFS in patients treated with a venetoclax-based treatment. Of interest, chemoimmunotherapy (but not venetoclax combinations) was associated with an increased risk of acquiring chromosomal aberrations at CLL relapse [20]. The most common grade ≥ 3 events were cytopenia and infections. In particular, severe infections were more common with chemoimmunotherapy (19%) and GIV (21%) than with VenR (11%) or VG (13%). Cardiac disorders were most frequent with GIV (FCR/BR 7%, VenR 8%, GV 8%, GIV 18%). The rate of second primary malignancies was higher with chemoimmunotherapy.

HOVON 139/GiVe was an open-label, randomized phase 2 trial performed in the Netherlands [21]. After 12 months of VG, patients were randomly assigned to receive venetoclax consolidation irrespective of MRD or only in patients with detectable MRD4 (i.e., ≥10^−4^). Seventy patients started VG, 62 were randomly assigned, 32 received 12 cycles of venetoclax consolidation and 30 venetoclax consolidation based on the MRD results. The median age was 72 years and 3% had TP53 abn. The ORR was 94% and the CR rate was 31%. Eighty-four percent of the patients achieved uMRD4 in the PB and 79% in the BM. Sixty-nine percent of the patients in the venetoclax consolidation group and 37% in the MRD-guided consolidation group had any adverse event. The most common grade 3 or higher adverse events were infection (6% in the consolidation subgroup and 3% in the MRD-guided consolidation subgroup) and neutropenia (6% and 7%, respectively). No treatment-related deaths occurred.

CAPTIVATE was a phase II study that investigated first-line ibrutinib plus venetoclax (I+V, 3 cycles of ibrutinib and then 12 cycles of combined I+V) in patients < 70 years of age in MRD-guided randomized discontinuation (MRD cohort) and fixed duration (FD cohort) cohorts [22]. Patients in the MRD cohort who reached uMRD4 were randomly assigned 1:1 to double-blind placebo or ibrutinib, while patients with detectable MRD4 (dMRD4, >10^−4^) randomly received open-label ibrutinib or I+V (1:1) [23]. Among 164 patients of the MRD cohort, after 12 cycles of I+V the best uMRD4 rates were 75% and 68% in the PB and in the BM, respectively. With a median follow-up of 31 months, in the uMRD4 subgroup the 1-year DFS rate was 95% with placebo and 100% with ibrutinib (p not significant). In the FD cohort, 159 patients were enrolled. The rate of complete remission was 55% and the best uMRD4 rates were 77% in the PB and 60% in the BM. The 24-month PFS and OS rates were 95% and 98%, respectively. Further analysis showed that deep and durable responses, as well as sustained PFS with fixed-duration I+V are maintained in patients with high-risk genomic features (such as U-IGHV, TP53 abn or CK) [24,25]. Adverse events were similar to those reported in other studies (the most common grade ≥ 3 adverse events were neutropenia (33%) and hypertension (6%)), were more common during the first 6 months of I+V and generally decreased over time.

FLAIR was a phase 3, multicenter, randomized, controlled trial comparing I+V, I+R and ibrutinib monotherapy with FCR [26]. So far, the comparisons of FCR vs. I+R and FCR vs. I+V have been reported. In the I+V arm, after 2 months of an ibrutinib 420 mg OD lead-in phase, venetoclax was added in a ramp-up phase followed by 400 mg once daily for at least 2 years or up to 6 years. The duration of I+V was double the time taken to achieve uMRD4 in the BM. In other words, if the patient needed 1 year to achieve uMRD4 then I+V will last 2 years and if they needed 3 years then I+V will last 6 years. Patients with 17p- were included if the deletion was present in <20% of CLL cells. A total of 523 patients were randomly assigned to the I+V or FCR, the median age was 62 years and 50% had U-IGHV and 10% harbored BCR stereotype #2. At a median of 44 months, the 3-year PFS was 97% vs. 77% for I+V vs. FCR, respectively (hazard ratio 0.13, *p* < 0.0001). In the subgroup analysis, the benefit of I+V over FCR was seen across all subgroups except among M-IGHV patients. The 3-year OS was 98% vs. 93% for I+V vs. FCR, respectively (hazard ratio 0.31, *p* < 0.005). The percentage of patients with uMRD4 in the BM after 2 years was 52% in the I+V arms but raised to 66% after 5 years. At 3 years, 58% of the patients in the I+V arm stopped therapy due to uMRD4. The most common grade 3–4 adverse events were cytopenia (anemia in the FCR arm and neutropenia in the I+V arm) and infections. Of note, while the risk of infection was similar between the I+V and FCR arms, the rate of cardiac severe adverse events was higher in the I+V group than in the FCR group. In a previous publication, authors reported an interim analysis of 386 patients assigned to I+R (ibrutinib given at standard dose according to the MRD result and up to 6 years) compared to 385 patients treated with FCR [27]. After a median follow-up of 53 months, the median PFS was not reached with I+R but was after 67 months with FCR (*p* < 0.0001). No difference in the OS was found. Eight (2%) and two (1%) sudden unexplained or cardiac deaths occurred in the I+R and FCR arms, respectively.

Sonrotoclax (previously known as BGB-11417) is a BH3 mimetic BCL2i with a higher potency than venetoclax in biochemical assays that was tested in combination with zanubrutinib in the BGB-11417-101 phase 1/1b dose-escalation/expansion study [28]. Patients received zanubrutinib (320 mg daily or 160 mg twice daily) for 2 or 3 months before starting sonrotoclax using a weekly ramp-up schedule starting from 1 mg to 160 mg or 320 mg QD. One hundred and seven patients were recruited, the median age was 62 years, 62% had U-IGHV and 26% had TP53 abn. The uMRD4 rate in the PB after 6 months was 48% and 78% with sonrotoclax 160 mg and 320 mg, respectively. At a median follow-up of 10 months, no patient had experienced disease progression or died. No TLS nor atrial fibrillation events occurred; infections of grade 3 or more occurred in 8% of patients; only 1% of the patients discontinued therapy.

For elderly and/or unfit patients, SEQUOIA was an open-label, global, multicenter, phase 3 trial that included a small nonrandomized cohort of patients with 17p deletion (Arm D) who were treated with zanubrutinib at standard dose for 3 months followed by zanubrutinib+venetoclax (Z+V) for 12–24 cycles until progressive disease, unacceptable toxicity or achievement of uMRD4 [29]. As of 1 June 2021 (data cutoff), 35 out of 80 planned patients with centrally confirmed TP53 deletion were enrolled. Thirty-two patients remained on study treatment and 3 (9%) discontinued treatment due to withdrawal of consent, disease progression or lung cancer. After a median follow-up of 10 months, the ORR was 97%, the CR rate was 14% and uMRD was achieved in 14% of patients. Thirteen patients (37%) had grade ≥ 3 adverse events, most frequently neutropenia (11%) and diarrhea (6%). No TLS events occurred.

The GLOW trial was a randomized multicenter phase 3 study performed in patients over 65 years old or who were younger with comorbidities (defined by a cumulative illness rating scale score of more than 6 or creatinine clearance less than 70 mL/min) [30]. The median age was 71 years and 7% had TP53 abn. Patients were randomized to I+V (the same schedule as that used in the Captivate study) or chlorambucil–obinutuzumab (G-CHL) groups. One hundred and six patients were assigned to I+V and 105 to G-CHL. The 42-month PFS rates were 75% vs. 25% for I+V vs. G-CHL, respectively (hazard ratio 0.26, *p* < 0.0001). The best uMRD rate in the BM via NGS was significantly higher for I+V (56%) than for G-CHL (21%; *p* < 0.001). Despite patients with U-IGHV status being able to achieve higher rates of uMRD than IGHV-mutated cases (61% vs. 40%), the proportion of patients with sustained uMRD was lower among U-IGHV cases (3-year uMRD 31% vs. 34%). Adverse events of grade 3 or higher occurred in 76% and 70% patients receiving I+V and G-CHL, respectively, with neutropenia (35% vs. 50%), infections (17% vs. 12%) and diarrhea (10% vs. 1%) being the most common. Overall, there were 15 (14%) and 30 (29%) deaths in the I+V and G-CHL arms, of which 3 (3%) and 10 (10%), respectively, were due to post-treatment infection.

In the phase 2 study by the MD Anderson Cancer Center, I+V combination was assessed in patients aged 65 years or older or with high-risk markers such as TP53 abn, 11q deletion or U-IGHV [31]. The I+V combination was administered for 24 cycles. In the original protocol, patients who remained BM dMRD4 at the EOT could continue ibrutinib monotherapy, then the trial protocol was amended to allow an additional 12 cycles of I+V. Eighty patients were enrolled, the median age was 65 years, 30% were 70 years old or older and 92% of the patients had U-IGHV, TP53 abn or 11q-. Sixty-six percent of the patients achieved BM uMRD4 remission after 24 cycles. Responses were seen across all high-risk subgroups. After a median follow-up of 39 months, the 3-year PFS and OS were 93% and 96%, respectively. No patient had CLL progression, but two cases of Richter syndrome transformation were documented. The adverse-event profile was similar to that reported with I+V, but three (4%) patients developed TLS.

The CLL14 phase 3 trial assessed 12 months of FD treatment of VG or G-CHL in patients with a CIRS score of at least 6 or a creatinine clearance of <70 mL/min [32]. Overall, 432 patients were enrolled; the median age was 72 years, the median CIRS score was 8 and 216 patients were randomized to each arm. After a median follow-up of 65 months, the PFS was significantly superior for VG compared to G-CHL; the 5-year PFS rates were 63% and 27%, respectively (hazard ratio 0.35, *p* < 0.0001). A lymph node size > 5 cm and TP53 abn were associated with an increased risk of relapse as well as of death, together with a complex karyotype (CK). In both arms, MRD status at the end of therapy was associated with a longer PFS. Detectable MRD4 status was associated with increased expression of the multi-drug resistance gene ABCB1 (MDR1), whereas uMRD6 (<10^−6^) was associated with increased expression of the BCL2L11 gene, which encodes the pro-apoptotic BIM protein. Inflammatory pathways were also enriched in dMRD4 patients. Severe neutropenia occurred in 53% of the patients but only 5% developed febrile neutropenia. Grade 3–4 infusion-related reactions and tumor lysis syndrome were recorded in 9% and 1% of the patients, respectively.

## 3. Fixed-Duration Double Therapies in Relapsed–Refractory Patients

In the relapsed–refractory setting, no study has so far reported on VG therapy, whereas I+V was evaluated in the VISION, IMPROVE and CLARITY clinical trials (Table 1).

HOVON141/VISION was a randomized phase 2 trial conducted in the north of Europe in patients who had not been exposed to BTKis or BCL2is [33]. A total of 108 patients out of 225 completed the I+V treatment and were tested for MRD at cycle 15; those with uMRD4 in the PB and BM were randomized 1:2 to receive ibrutinib (*n* = 24) or treatment cessation (*n* = 48). Patients who were dMRD4 also received ibrutinib. Patients who became dMRD2 (i.e., >10^−2^) during the follow-up restarted therapy with I+V. The remaining 153 out of 225 patients continued with ibrutinib monotherapy. The trial was updated at the 2023 EHA congress, showing that at 51 months, PFS was 92% vs. 81% vs. 75% and time to next treatment was 8% vs. 4% vs. 12% in patients randomized to ibrutinib, treatment cessation and those not randomized, respectively [34]. Among uMRD4 patients randomized to ibrutinib maintenance, 33% were still uMRD4 and 54% discontinued treatment, while among patients randomized to observation, only 21% were still uMRD4 and 17% went off the trial. At the last follow-up, 19 (8%) patients, mainly with TP53 abn or CK, restarted I+V. Infection (58% any grade and 28% grade 3 or above), neutropenia (40% any grade and 35% grade 3 or higher) and gastrointestinal (24% any grade and 6% grade 3) adverse events were the most frequently reported. One (0.4%) patient who was not randomly assigned had a fatal bleeding.

CLARITY was a phase 2 trial conducted in the United Kingdom that combined I+V in patients with relapsed or refractory CLL [35]. Patients previously treated with BTKis or BCL2is were excluded. The schedule of I+V was the same as that used in the FLAIR study. The duration of therapy was scheduled as follows: uMRD4 in both PB and BM at month 8 to stop I+V at month 14, dMRD4 at month 8 but uMRD4 in both PB and BM at month 14 and/or at month 26 to stop I+V at month 26, and dMRD4 at month 26 to stop venetoclax but continue ibrutinib until progression. Fifty-four patients were enrolled; the median age was 64 years (range 31–83 years), 74% displayed U-IGHV, 22% 17p- and the median number of prior therapies was one (range 1–6). At month 14, 53% and 36% of the patients achieved uMRD in the PB and BM, respectively, which increased to 67% and 44%, respectively, at month 26. After a median follow-up of 21 months, one patient with BCR subset #2 relapsed but no patients died. A single case of biochemical TLS was observed. Of note, 63% of the patients developed grade 3 or higher neutropenia and infective events occurred in 17% of the patients.

The IMPROVE trial adopted another strategy, starting with venetoclax for 12 months and discontinuing venetoclax if patients were uMRD4 otheriwse combining ibrutinib till month 24 or uMRD4 [36]. Patients still dMRD4 after month 24 continued ibrutinib. Thirty-eight patients (29% with TP53 abn; 79% with U-IGHV) started venetoclax, 45% with uMRD4 stopped venetoclax and 55% added ibrutinib. With I+V, 84% of the patients achieved uMRD4, thus stopping both drugs. Only two (5%) continued ibrutinib maintenance. After a median follow-up of 37 months, 22% of the patients remained uMRD4 and 26% progressed, including 10% who restarted venetoclax. Neutropenia was the most frequent grade 3–4 adverse event.

In addition, retreatment with I+V is allowed in the CAPTIVATE study and preliminary results are encouraging [25]. ReVenG will assess the efficacy and safety of VG in patients who initially responded to first-line VG for at least 12 months [37].

## 4. Triplet Combinations

Triplet therapy combinations involving a BTKi, a BCL2 inhibitor and CD20-targeting mAbs represent a promising therapeutic strategy for CLL still under investigation, especially for those patients harboring high-risk markers such as TP53 abn or CK, in which it is hard to achieve sustained uMRD4 remission. Despite this, only a few randomized trials are available (Table 2).

The triplet combination of ibrutinib, venetoclax and obinutuzumab (IVO, also known as GIV regimen) previously demonstrated an uMRD4 rate of 67% in treatment-naïve (TN) patients and 50% in relapsed/refractory (RR) patients after 14 cycles of therapy in the first phase 2 study presented by Rogers et al. [38].

The CLL2-GIVe study [39], an open-label, single-arm, multicenter trial, explored this triple combination in CLL patients with TP53 abn. Patients received six cycles of obinutuzumab and ibrutinib at standard dose from cycle 1 day 1, while the venetoclax ramp-up phase started from cycle 1 day 22, followed by a consolidation phase with I+V for six additional cycles and then the ibrutinib single agent till cycle 15. For patients not achieving CR uMRD4 at cycle 15, ibrutinib was prolonged until cycle 36. Forty-one patients were enrolled. The ORR was 100%, including 59% CRs. Seventy-eight percent of patients achieved uMRD4 in PB and 66% in the BM; this rose to 96% and 88%, respectively, in the CR subpopulation [39]. The 36-month PFS and OS were 80% and 93%, respectively. The authors showed that patients harboring both 17p- and a TP53 mutation had a shorter PFS compared with patients who had only a TP53 mutation. Twenty-four percent of patients registered adverse events grade ≥ 3, mainly neutropenia (49%) and infections (20%). The incidence of adverse events was higher during the induction phase but decreased over time. Overall, 83% of adverse events led to a dose reduction and 14% caused drug discontinuation. Atrial fibrillation and hypertension occurred at rates of 2.4% between cycles 1 and 12 and 4.9% (between cycles 1 and 6), respectively.

This triple combination was also explored in the previously mentioned CLL13 trial, confirming very high and deep remission in patients without TP53 abn that was balanced by higher rates of infections and cardiovascular events [19].

The combination of acalabrutinib (A), venetoclax (V) and obinutuzumab (AVO) demonstrated high activity in the frontline setting, as previously reported by Davids M. et al. in a single-arm phase 2 trial. The study did not met the primary end point, due to a lower rate of uMRD4 in the BM (38%) [40]. The study was amended, enrolling 68 patients, i.e., the previously reported 37 patients and 31 additional patients with TP53 abn. Patients received acalabrutinib 100 mg twice a day for 28 days, followed by six cycles of acalabrutinib plus obinutuzumab. Venetoclax was introduced in cycle 4 using an accelerated ramp-up phase (20 mg C4D1, 50 mg C4D2, 100 mg C4D8, 200 mg C4D15 and 400 mg C4D22). AV treatment was discontinued at cycle 15 or cycle 25 in patients reaching CR and uMRD4 in the BM, otherwise they continued AV until disease progression or unacceptable toxicity. After a median follow-up of 35 months, the best overall response rate was 98%, the CR rate was 48% and high rates of uMRD4 (PB 86% and BM 86%) were observed. Similarly, in the TP53-aberrant population, the CR rate was 52% and the uMRD4 rate was 83% in the BM. The 3-year PFS and OS were 93% and 99%, respectively [41]. Regarding safety, 21% of patients required dose reductions but no one discontinued treatment due to adverse events. Most adverse events, such as headache, fatigue and nausea, were of grade 1–2. Grade ≥ 3 neutropenia occurred in 37% of patients and thrombocytopenia in 28%, but there were no major bleeding events. Nine percent had COVID-19, including one death. As regard cardiovascular toxicity, hypertension occurred in 27% of patients and 3% of patients reported atrial fibrillation, but there were no cases of ventricular arrhythmia [41].

The second-generation BTKi zanubrutinib was explored in combination with venetoclax and obinutuzumab as an initial therapy for CLL/SLL in the BOVen phase 2 trial [42]. Patients received 160 mg zanubrutinib daily and obinutuzumab at a standard dose from cycle 1 to 8; venetoclax was given starting from cycle 3, the typical ramp-up phase. Treatment was given for a minimum of 8 months and up to 24 months in an MRD-driven strategy. Fifty-two patients were enrolled; the median age of the patients was 62 years (range 23–77 years), while 72% were U-IGHV and 13% harbored TP53 abn. After a median follow-up of 40 months, the ORR was 100%, including 55 CRs, but the median PFS was not reached. Ninety-five percent of the patients had uMRD4 in the PB and 89% in the BM. The median MRD-free survival was 30 months. In this study, a potential new surrogate end point was also explored. ΔMRD400 was measured at cycle 5 and defined as a reduction in MRD of at least 400-fold from baseline. Notably, for twenty-one patients (60%) who achieved ΔMRD400, the median MRD-free survival was not reached vs. 18 months in those who did not achieve ΔMRD400 (*p* = 0.003). In addition, ΔMRD400 at cycle 5 predicted early uMRD4 in the BM [43]. During treatment, cytopenia was common, but only of grade 1–2, while grade ≥ 3 neutropenia occurred in 23% of patients and thrombocytopenia in 8%. Grade ≥ 3 pneumonia and infusion-related reactions were registered in 6% and 4% of patients, respectively.

In the Phase 2 study from the German CLL Study Group, the AVO combination was explored in relapsed/refractory patients after an optional bendamustine debulking [44]. Patients started obinutuzumab at cycle 1, A at cycle 2 and V at cycle 3, followed by a maintenance phase from cycle 10 with AV unchanged and obinutuzumab administered every 3 months up to 24 months or until CR and uMRD4 in PB in two consecutive assessments. Forty-five patients were enrolled, 47% had already received a targeted agent and 32% had TP53 abn. Seventy-six percent of the patients had uMRD4 in the PB after 6 months of the triple therapy. At the last available follow-up, 71% of the patients started maintenance and 28% were able to stop treatment. After a median follow-up of 14 months, no progressions nor deaths were observed and two patients developed Richter syndrome transformations. The most common adverse events of grade 3 or higher during the entire treatment were thrombocytopenia and neutropenia (27%, each), TLS (11%), infections (11%), infusion-associated reactions (9%) and anemia (9%).

## 5. CAR T Cells

Patients with relapsed-refractory CLL who experience disease progression after both a BTKi and a BCL2i have poor outcomes and no established standard of care, indicating a critical unmet clinical need. In previous studies, CD19-directed chimeric antigen receptor (CAR) T-cell therapies showed promise as a potential treatment modality for relapsed/refractory CLL [45,46,47,48,49]; however, T cells from patients with CLL are known to be dysfunctional and producing functional CAR T cells capable of both in vivo expansion and persistence can be challenging [50]. CAR T cells mediate HLA-unrestricted tumor cell killing by enabling T cells to bind target cell surface antigens. Upon engagement, CAR T cells form a non-classical immune synapse that is required for their effector function and trigger the release of perforin and granzyme, engagement of the Fas axis and the release of cytokines [51].

In this context, lisocabtagene maraleucel (liso-cel) is a second-generation autologous, CD19-directed, CAR T-cell product administered at equal target doses of CD8+ and CD4+ CAR T cells. Furthermore, the early selection of CD8+ and CD4+ T cells and removal of non-T-cell impurities before activation and transduction of the CAR T cells increased T cell purity. Additionally, this removal of residual CD19+ CLL cells before transduction decreased the risk of inhibiting CAR T-cell expansion and prevented off-target transduction of B cells. The TRANSCEND CLL 004 study is a single-arm, phase 1–2 study conducted in CLL patients previously treated with at least two lines of therapy, including a BTKi, and who received liso-cel at one of two different doses: 50 × 10^6^ (cohort 1) or 100 × 10^6^ (cohort 2) CAR T cells [52] (Table 3). One hundred and thirty-seven enrolled patients underwent leukapheresis but only 117 received Liso-cell (97 were efficacy evaluable, 9 from cohort 1 and 88 from cohort 2). The median age was 65 years (range 49–82), 42% had TP53 abn, 47% U-IGHV, 61% CK and the median number of lines of prior therapy was five (range 2–14, including 70/117 double-refractory patients). In cohort 2, the ORR was 44%, including 20% CRs; the uMRD rate was 64% in PB and 60% in BM. With a median follow-up of 24 months, the median PFS and OS were 12 and 30 months, respectively. The rate of cytokine release syndrome (CRS) was 85% (mainly grade 1–2; 8% grade 3), the rate of neurological events (ICANS) was 45% (18% grade 3 and 1% grade 4), the rate of prolonged grade 3 cytopenia was 54% and the rate of grade ≥ 3 infections was 18%. Sixty-nine percent of the patients received tocilizumab and/or corticosteroids for CRS or neurological events [53].

## 6. Bispecific Antibodies

Bispecific antibodies (BsAbs) represent a new class of immunotherapeutic drugs; they are engineered to recognize and bind two different antigens [54]. There are two major classes that are categorized based on the presence or the lack of an Fc region. In the first one, BsAbs are similar to IgGs and determine Fc-mediated effector functions such as antibody-dependent cell cytotoxicity, complement-dependent cytotoxicity and antibody-dependent cell phagocytosis. Conversely, BsAbs without an Fc region are small molecules composed of whole-fragment antigen-binding (Fab) regions or only some fragment of the Fab connected with a linker peptide. BsAbs are small molecules that spread well into tissues but are also subjected to intracellular degradation and renal elimination. T-cell engagers are the most frequently used class of BsAbs and act via targeting tumor-associated antigens using one Fab arm and a T cell using the other one. The rationale behind the use of T-cell engagers is to activate and direct the patients’ immune system cells towards the tumor cells. The immune synapsis generated by BaAbs induces polyclonal T-cell activation independent from TCR specificity, human leukocyte antigen (HLA) and co-stimulatory signals. Once activated, cytotoxic T lymphocytes release perforins and granzymes, causing the lysis of the target cells. Afterwards, multiple cytokines such as interleukin (IL)-2, IL-6, IL-10, interferon-gamma (IFN-γ) and tumor necrosis factor-alpha are secreted, leading to the activation of other immune cells, such as B cells, macrophages and NK cells [54].

The use of T-cell engagers in hematological malignancies started with the use of the anti CD3 x CD19 bispecific antibody blinatumomab, which is currently approved in the treatment of acute lymphoblastic leukemia. Some of these new molecules are also being tested in CLL patients, even if most of the available data came from preclinical studies. A study conducted in 2013 showed that blinatumomab stimulates the autologous T-cell killing of CLL cells in vitro, suggesting a possible clinical benefit [55]. Interestingly, blinatumomab was instead tested in nine patients with Richter syndrome transformation and, after two cycles, only two (22%) patients responded [56].

Since then, other molecules have currently been under investigation in other settings such as multiple myeloma and aggressive lymphomas. Most of the other BsAbs that have been studied in lymphoproliferative disorders target CD20 and CD3.

Mosunetuzumab is a full-length, fully humanized IgG1 anti-CD20/CD3 bispecific antibody that is approved for the treatment of R/R follicular lymphoma [57]. A phase I/II study is currently ongoing to evaluate the CD3 x CD20 antibody mosumetuzumab in CLL as a single agent and in combination with atezolizumab (anti-PD-L1); however, to date, no results are available. Fixed-duration mosunetuzumab monotherapy demonstrated activity with durable responses (ORR 40%, CR 20% and duration of CR ≥ 20 months at data cut-off) and a manageable safety profile in 20 patients with R/R Richter syndrome transformation, as presented at the annual ASH meeting 2023 [58].

Another antibody that is currently being tested in CLL is plamotamab, a humanized bispecific antibody that binds both CD20 and CD3. In a phase I/II dose-escalation trial, plamotamab showed a good safety profile and evidence of clinical activity in heavily pretreated NHL, including CLL patients [59].

In the EPCORE CLL-1 trial, epcoritamab (EPCO), a full-length anti-CD20 x CD3 IgG1 T-cell engager, showed a good safety profile and antitumor activity in heavily treated patients with high-risk R/R CLL and Richter syndrome [60]. Patients received EPCO subcutaneously at 48 mg once a week in cycles 1–3, every two weeks in cycles 4–9 and then every 4 weeks until disease progression. Updated data were presented at the iwCLL 2023 meeting on 23 heavily pre-treated patients who had a median number of previous therapies of four (range 2–10), all of whom were previously exposed to a BTKi and 83% also to a BCL2i [61]. The ORR was 62%, including 33% CRs and 75% with uMRD4. The median time to response was 2 months and the estimated 9-month PFS was 67%. Almost all patients experienced CRS, including 17% events of grade 3, but all cases resolved. CRS events were more common when patients received the first full dose of EPCO at cycle 1 day 15. Cytopenia were recorded in half of the patients and ICANS were occurred in 13% (all of grade 1 or 2). In addition, in the EPCORE CLL-1 expansion cohorts and new escalations, the efficacy of EPCO will be evaluated as a single agent and in combination with venetoclax in CLL and with lenalidomide or R-CHOP in patients with Richter syndrome [62].

Glofitamab is another CD20 x CD3 BsAbs that is able to bind B cells with two high-avidity sites and a T cell with one site; it is approved for relapsed–refractory large B cells lymphoma. Glofitamab was administered intravenously for 12 cycles at 25 mg. Data on 11 patients with Richter syndrome transformation were presented at the 2023 ICML (International Conference on Malignant Lymphoma) congress; the median age of the patients was 71 years and all were refractory from the most recent therapy [63]. After a median follow-up of 41 months, the best ORR was 64%, including a 46% CR rate. Eighty percent (4/5) of patients with a CR remained in remission at the data cut-off. CRS events occurred in 73% of the patients, mainly during cycle 1, but only 18% of these events were grade 3 or higher. Half of the patients received tocilizumab for CRS management. ICANS were recorded in 18% of the patients, 9% of which were grade 3 or 4.

MGD011 is another type of CD19 x CD3 bispecific protein with an engineered Fc domain that displays antitumor activity in several in vitro and in vivo models of B-cell malignancies. MGD011-mediated B-cell killing was accompanied by target-dependent T-cell activation and expansion, cytokine release and upregulation of perforin and granzyme B, which translates to the CLL in a partial restoring of the T cell immunological dysfunctions [64].

Another interesting possibility is to improve the efficacy of anti CD20/CD3 T-cell engagers by combining with BTKis or BCL2is, as suggested in a study conducted by Mhibik and colleagues in which EPCO was evaluated in vitro and in vivo models with a BTKi and venetoclax [65].

The data listed in Table 3 support the investigation of the use of T-cell engagers in combination with other established CLL treatments to consolidate responses or to overcome drug resistance.

## 7. Discussion

The clinical heterogeneity that characterizes CLL is largely due to the age at the start of treatment and to the wide spectrum of comorbidities in patients, ranging from young and fit patients to very elderly patients suffering from several other diseases, as well as the variability of genetic aberrations within CLL cells [66,67,68]. TP53 abn, IGHV gene mutational status and karyotype represent the main prognostic factors of CLL [9,69,70]. Also, in the era of novel target agents, TP53 abn remains a negative predictive marker for CLL and is associated with worse PFS and OS, especially for those cases harboring both 17p- and TP53 mutations [71,72,73,74]. While a remarkable rate of U-IGHV patients are able to reach uMRD4, some of them relapse earlier than M-IGHV patients after FD therapy.

Although a large number of studies on novel combinations and newer strategical options for CLL therapy have been recently published or presented at international conferences, many questions remain open. For example, what are the best “end points” for clinical investigational studies and whether the weight of these variables has a direct effect on the management of CLL patients in daily care settings. Furthermore, the real meaning of uMRD in clinical practice is unknown, as is if all patients have to reach it. On the other hand, the risk of cardiovascular events, sudden deaths and the unsustainable costs associated with continuous BTKi treatment are well known. Mutations of BTK at the ATP-binding site Cys481 (e.g., C481S, C481Y, C481R and C481G, which account for 70–90% of all the mutations) and mutations at PLCG2 are the main mechanisms of resistance to covalent BTKis described [13,75]. Recently, non-C481 BTK, namely “gatekeeper mutation” (i.e., T474x) and kinase-dead mutations (i.e., L528x) have been reported, both after zanubrutinib and acalabrutinib. These non-C481 BTK mutations decrease the activity of the non-covalent BTKi pirtobrutinib [76,77].

The combination of a BTKi and a BCL2i is a very appealing fixed-duration treatment strategy. BTKis and BCL2is have distinct and complementary mechanisms of action that work synergistically. While BCL2is are able to kill both resting and dividing CLL cells, BTKis can mobilize cells out of lymph nodes and other secondary lymphoid organs, thus priming CLL cells to BCL2i-induced killing. Furthermore, this completely oral therapy is more manageable than using combinations with intravenous drugs, such as obinutuzumab, mitigating the risk of TLS and severe infusion reactions.

uMRD is generally considered as a predictor of long-term outcome in CLL, irrespective of age, class of risk, disease phase and type of therapy. In this context, clinical trials assessing MRD after the combination of a BTKi + a BCL2i suggest that most patients are able to reach uMRD4 and experience long-lasting remission. However, some patients with dMRD4 at the end of therapy and those with unfavorable genetic markers (TP53 abn, CK, U-IGHV) display a shorter MRD doubling time and likely early relapse. Thus, an MRD-guided treatment cessation and MRD-based re-initiation of targeted therapy seems to be more feasible for patients with CLL, who are usually elderly and suffer from various comorbidities. This sequential MRD-guided approach allows identification of the few patients benefiting from continuous therapy or maybe a consolidation with BsAbs or CAR T cells to clear the neoplastic clone.

In addition, it is not clear what the role of triple therapies in CLL will be, i.e., an anti-CD20 monoclonal antibody in the BCL2i–BTKi treatment, since the results in both naive and relapse/refractory patients are not much better than those derived from double combinations. Another consideration should be performed regarding BsAbs or CAR T cells that are likely not to be cost-effective in CLL [78] and not available in most low-income countries. In the future, BsAbs might be tested in combination with BTKis and/or BCL2is, whose activity might be higher than triple combinations with monoclonal antibodies. A further approach to try to improve CAR-based strategies in CLL is the use of off-the-shelf products that involve using T or NK cells from healthy donors. Other advantages of this approach include the potential consistent reduction of manufacturing labor and costs and the possibility to treat more patients with the same product.

## 8. Conclusions

Combinations of a BTKi plus venetoclax as a fixed-duration therapy will be the standard of care for most patients with CLL, while continuous therapy will be likely limited only to high-risk and/or unfit patients. In the future, second-generation (acalabrutinib, zanubrutinib), non-covanlent BTKis (pirtobrutinib, nemtabrutinib) [79] and BTK degraders (BGB-16673, NX-2127, NX-5948) [14] (Figure 1), also known as chimeric degradation activation compounds, will be tested in combination with venetoclax but also second-generation BCL2is (sonrotoclax, lisaftoclax) with an MRD-driven approach, further improving the treatment landscape of CLL.

## Figures and Tables

**Table 1 cancers-16-01290-t001:** Double combinations trials in CLL.

Line	Study	Treatment	Phase	Comparator	N	Age	TP53 abn	ORR/CR	PFS	OS	uMRD (10^−4^)	Median FUP (Months)
** *Frontline* ** ** *Young Patients* **	** *VERITAS* **	Venetoclax, Rituximab	2	Single arm	75	54	12%	95%/76%	3-yy 96%	3-yy 96%	69% PB 59% BM	33
	** *HOVON 139* **	Venetoclax, Obinutuzumab	2	single arm	231	72	13%	94%/31%	3-yy 85%	3-yy 94%	84% PB 79% BM	35
	** *CAPTIVATE* **	Ibrutinib, Venetoclax	2	MRD arm FD arm	164 159	58 60	20% 17%	97%/46% 96%/55%	2-yy 96% 2-yy 95%	2-yy 98% 2-yy 98%	75% PB 68% BM 77% PB 60% BM	31 28
	** *CLL13* **	Venetoclax, Obinutuzumab	3	FCR/BR venetoclax, rituximab	229	61	0%	100%/59%	3-yy 88%	3-yy 96%	86% PB 73% M	39
	** *FLAIR* **	ibrutinib, Rituximab Ibrutinib, Venetoclax	3	FCR	386 260	63 62	1% 0.4%	91%/21% 84%/71%	3-yy 90% 3-yy 97%	3-yy 95% 3-yy 98%	4% PB 4% BM 71% PB 52% BM	53 44
	** *BGB-101* **	Zanubrutinib, Sonrotoclax	1/2	Single arm	197	62	26%	100%/32%	1-y 100%	1-y 100%	78% PB	10
** *Frontline* ** ** *Elderly Patients* **	** *SEQUOIA D* **	Zanubrutinib, Venetoclax	2	Single arm	49	65	100%	97%/14%	1-y 95%	1-y 95%	14% PB	12
	** *MDACC* **	Ibrutinib, Venetoclax	2	Single arm	80	65	23%	100%/86%	3-yy 93%	3-yy 96%	PB n.r. 66% BM	39
	** *GLOW* **	Ibrutinib, Venetoclax	3	Chlorambucil, Obinutuzumab	106	71	7%	87%/39%	3.5-yy 75%	3.5-yy 87%	81% PB 56% BM	57
	** *CLL14* **	Venetoclax, Obinutuzumab	3	Chlorambucil, Obinutuzumab	216	72	12%	85%/50%	5-yy 63%	5-yr 87%	76% PB 57% BM	65
** *Relapsed Refractory* **	** *VISION* **	Ibrutinib, Venetoclax then Ibrutinib or stop therapy	2	Single arm	225	68	24%	85%/64%	3-yy 88%	3-yy 94%	50% PB 37% BM	51
	** *CLARITY* **	Ibrutinib, Venetoclax	2	Single arm	54	64	22%	89%/51%	2-yy 98%	2-yy 100%	67% PB 44% BM	21
	** *IMPROVE* **	Venetoclax, Ibrutinib	2	Single arm	38	64	29%	95%/53%	3-yy 75%	3-yy 95%	92% PB 87% BM	36

N, number of patients enrolled in the study; ORR, overall response rate; CR, complete response; PFS, progression-free survival; OS, overall survival; uMRD, undetectable measurable residual disease; PB, peripheral blood; BM, bone marrow; FUP, follow-up, n.r. = not reported, y = year, yy = years.

**Table 2 cancers-16-01290-t002:** Triplet combination trials in CLL.

	Study	Treatment	Phase	Comparator	N	Age	TP53 abn	ORR/CR	PFS	OS	uMRD (10^−4^)	Median FUP (Months)
** *Frontline* **	** *IVO* **	Ibrutinib, Venetoclax, Obinutuzumab	2	Single arm	25	59	12%	84%/32%	2-yy 96%	2-yy 96%	PB n.r. 67% BM	24
	** *CLL2-GIVe* **	Ibrutinib, Venetoclax, Obinutuzumab	2	Single arm	41	62	100%	100%/59%	3-yy 80%	3-yy 93%	78% PB 66% BM	36
	** *CLL13* **	Ibrutinib, Venetoclax, Obinutuzumab	3	venetoclax rituximab, FCR/BR	231	61	0%	94%/62%	3-yy 91%	3-yy 95%	92% PB 78% BM	39
	** *AVO* **	Acalabrutinib, Venetoclax, Obinutuzumab	2	Single arm	68	63	46%	98%/48%	3-yy 93%	1-y 99%	86% PB 86% BM	35
	** *BOVen* **	Zanubrutinib, Venetoclax, Obinutuzumab	2	Single arm	39	62	100%	100%/57%	n.r.	n.r.	92% PB 84% BM	14
** *Relapse* **	** *IVO* **	Ibrutinib, Venetoclax, Obinutuzumab	2	Single arm	25	58	4%	88%/24%	2-yy 8%	2-y 100%	PB n.r. 50% BM	24
	** *AVO* **	Acalabrutinib, Venetoclax, Obinutuzumab	2	single arm	45	60	32%	100%/18%	1-y 94%	1-yy 100%	76% PB 16% BM	26

N, number of patients enrolled in the study; ORR, overall response rate; CR, complete response; PFS, progression-free survival; OS, overall survival; uMRD, undetectable measurable residual disease; PB, peripheral blood; BM, bone marrow; FUP, follow-up, n.r. = not reported, y = year, yy = years.

**Table 3 cancers-16-01290-t003:** Chimeric antigen receptor (CAR) T cells and bispecific antibodies under investigation in CLL.

Drug	Bispecific Antibody	Phase	Development	CLL Setting	Administration	Administration Route	Treatment Duration
** *Liso-cel* **	CD19 CAR T	1/2	B-cell malignancies	Relapsed/Refractory	Single agent	Intravenous	single dose
** *Axi-cel* **	CD19 CAR T	1/2	B-cell malignancies	Relapsed/Refractory	Single agent	Intravenous	single dose
** *Tisa-cel* **	CD19 CAR T	1/2	B-cell malignancies	Relapsed/Refractory	Single agent	Intravenous	single dose
** *Brexu-cel* **	CD19 CAR T	1/2	B-cell malignancies	Relapsed/Refractory	Single agent	Intravenous	single dose
** *MGD011* **	CD20 x CD3	preclinical activity	B-cell malignancies	Relapsed/Refractory	Single agent	Intravenous	dose dependent
** *Plamotamab* **	CD20 x CD3	1	B-cell malignancies	Relapsed/Refractory Richter Syndrome	Single agent	Intravenous	until progression disease
** *GB261* **	CD20 x CD3	1/2	B-cell malignancies	Relapsed/Refractory	Single agent	Intravenous	until progression disease
** *Epcoritamab* **	CD20 x CD3	1/2	B-cell malignancies	Relapsed/Refractory Richter Syndrome	Single agent	subcutaneous	until progression or unacceptable toxicity
** *Mosunetuzumab* **	CD20 x CD3	1/2	B-cell malignancies	Relapsed/Refractory	Single agent or with atezolizumab	Intravenous or subcutaneous	until progression or unacceptable toxicity
** *Mosunetuzumab* **	CD20 x CD3	1/2	B-cell malignancies	Richter Syndrome	Single agent	Intravenous	fixed duration
** *Glofitamab* **	CD20 x CD3	1/2	B-cell malignancies	Richter Syndrome	Single agent	Intravenous	fixed duration

Axi-cel = axicabtagene ciloleucel; Brexu-cel = brexucabtagene autoleucel; Liso-cel = lisocabtagene maraleucel; Tisa-cel = tisagenlecleucel, CLL = chronic lymphocytic leukemia.

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
