# Peer review of "Innovative Combinations, Cellular Therapies and Bispecific Antibodies for Chronic Lymphocytic Leukemia: A Narrative Review"

_cancers, 2024, doi:10.3390/cancers16071290_

Round 1
Reviewer 1 Report
Comments and Suggestions for Authors
Summary: The manuscript by Visentin et al. provides a comprehensive overview of innovative treatment strategies for chronic lymphocytic leukemia (CLL), focusing on novel combinations, cellular therapies, and bispecific antibodies. The paper aims to highlight the evolving landscape of CLL treatment options, emphasizing the potential of these innovative approaches to improve patient outcomes. The main contributions of the paper lie in its thorough examination of current and emerging therapies for CLL, offering valuable insights into the future direction of CLL management. The manuscript's strengths include its clear organization, in-depth analysis of key molecular targets, and discussion of challenges and opportunities in CLL therapy.
General comments: Overall, the manuscript effectively covers various topics related to innovative therapies for CLL. The review topic is highly relevant given the rapidly evolving landscape of CLL treatment options, and the authors have done a commendable job in summarizing the current state of the field. One notable strength of the manuscript is the identification of gaps in knowledge, particularly in the discussion of challenges faced in the management of CLL, such as in elderly patients with comorbidities. The references cited are appropriate and support the arguments presented in the paper. However, it would be beneficial for the authors to provide more detailed insights into the potential limitations of the reviewed therapies and to discuss the conflicting evidence in the field.
Specific comments:
- Major English edits are needed. The language might not be the author’s first language, but it is completely understandable. Extensive edits by a professional are required, and the English used is incorrect and sometimes difficult to follow.
- Please make the title Sentence case. There are too many acronyms that can be minimized.
- In Figure 1, including a legend that clearly explains the abbreviations used for the targeted therapies mentioned would be helpful. The legend needs to be more detailed and needs to be abbreviated.
- CAR T-cells Section: Please clarify the specific mechanisms of action of CAR T-cell therapies in CLL, as this is briefly mentioned but needs to be elaborated upon.
- Include a table summarising the ongoing trials as mentioned in the text. Also, recommend including another table with additional information on the patient populations studied and the key outcomes reported in the Phase II/III trials with available results.
- Bispecific antibodies in CLL could be strengthened with specific examples or data from clinical studies.
- In general, for every study cited, please include survival (PFS, OS), response rate (ORR), phase (if clinical trial), number of patients (N), and p-value/ confidence interval, wherever applicable.
- Discussion: Expand on the potential challenges associated with the widespread adoption of cellular therapies in CLL, particularly regarding accessibility and cost-effectiveness.
- As part of the discussion, could you add potential implications of the findings presented in the review on clinical practice and a separate para on future directions for research in CLL therapy?
- Discussion: Consider removing the numbering in the paragraph: “Although a large number of data on novel combinations and newer strategical operations for CLL therapy have been recently published or presented at international conferences, many questions remain open:..”
- Draft a conclusion paragraph to summarize the key takeaways from the review and emphasize the significance of the discussed innovative therapies for improving outcomes in CLL patients.
Overall, the manuscript provides a valuable overview of innovative treatment approaches for CLL, but further elaboration on certain points and addressing the specific comments raised would enhance the clarity and depth of the review.
Comments on the Quality of English LanguageExtensive edits by a professional are required, and the English used is incorrect and sometimes difficult to follow.
Author Response
Dears Editor and reviewers,
thank you for your letter on 04 Mar 2023 informing that our manuscript was found of interest, pending the resolution of some minor issues.
Our paper was modified according to your and the reviewer suggestions as followed reported.
REVIEWER #1
Q1. Summary: The manuscript by Visentin et al. provides a comprehensive overview of innovative treatment strategies for chronic lymphocytic leukemia (CLL), focusing on novel combinations, cellular therapies, and bispecific antibodies. The paper aims to highlight the evolving landscape of CLL treatment options, emphasizing the potential of these innovative approaches to improve patient outcomes. The main contributions of the paper lie in its thorough examination of current and emerging therapies for CLL, offering valuable insights into the future direction of CLL management. The manuscript's strengths include its clear organization, in-depth analysis of key molecular targets, and discussion of challenges and opportunities in CLL therapy.
General comments: Overall, the manuscript effectively covers various topics related to innovative therapies for CLL. The review topic is highly relevant given the rapidly evolving landscape of CLL treatment options, and the authors have done a commendable job in summarizing the current state of the field. One notable strength of the manuscript is the identification of gaps in knowledge, particularly in the discussion of challenges faced in the management of CLL, such as in elderly patients with comorbidities. The references cited are appropriate and support the arguments presented in the paper. However, it would be beneficial for the authors to provide more detailed insights into the potential limitations of the reviewed therapies and to discuss the conflicting evidence in the field.
A1. We really appreciate the reviewer’s consideration. As suggested we added the limitations of the therapies.
Q2. Major English edits are needed. The language might not be the author’s first language, but it is completely understandable. Extensive edits by a professional are required, and the English used is incorrect and sometimes difficult to follow.
A2. Following the reviewer suggestion, the manuscript was revised by an English mother langue.
Q3Please make the title Sentence case. There are too many acronyms that can be minimized.
A3. The tile was corrected and only essential acronyms were used.
Q4. In Figure 1, including a legend that clearly explains the abbreviations used for the targeted therapies mentioned would be helpful. The legend needs to be more detailed and needs to be abbreviated.
A4. We abbreviated the legend to Figure 1 and we explained all the abbreviations.
Q5. CAR T-cells Section: Please clarify the specific mechanisms of action of CAR T-cell therapies in CLL, as this is briefly mentioned but needs to be elaborated upon.
A5. Mechanisms of action of CAR T-cells is reported on lines 417-420.
Q6. Include a table summarizing the ongoing trials as mentioned in the text. Also, recommend including another table with additional information on the patient populations studied and the key outcomes reported in the Phase II/III trials with available results.
Bispecific antibodies in CLL could be strengthened with specific examples or data from clinical studies.
In general, for every study cited, please include survival (PFS, OS), response rate (ORR), phase (if clinical trial), number of patients (N), and p-value/ confidence interval, wherever applicable.
A6. We apologize with the reviewer but likely there was an informatic issue, as in the submission there were 3 tables summarizing all the analyzed clinical trials. Please see tables 1-2-3.
Data on efficacy of mosunetuzumab, epcoritamab and globifatamab were added in the manuscript on lines 478, 686-688, 700-702.
For each trials data of the number of patients, ORR, CR, PFS, OS were added in Tables number 1 and 2.
Q7. Discussion: Expand on the potential challenges associated with the widespread adoption of cellular therapies in CLL, particularly regarding accessibility and cost-effectiveness.
As part of the discussion, could you add potential implications of the findings presented in the review on clinical practice and a separate para on future directions for research in CLL therapy?
Discussion: Consider removing the numbering in the paragraph: “Although a large number of data on novel combinations and newer strategical operations for CLL therapy have been recently published or presented at international conferences, many questions remain open:..”
Draft a conclusion paragraph to summarize the key takeaways from the review and emphasize the significance of the discussed innovative therapies for improving outcomes in CLL patients.
A7. Following the reviewer’s suggestion were rephrased the indicated paragraph, added limitations of the reported therapies and discussed accessibility and sustainability of new drugs in the discussion section.
Reviewer 2 Report
Comments and Suggestions for Authors
This review properly summarizes part of the actual knowledges derived from clinical trials testing novel combinations and/or novel agents.
Overall, this review allows to gain a proper overview of the various therapeutic strategies under investigation, the associated side effects, and the subsets of patients most likely taking advantage from the described therapies. Hence, it is likely to address a broad public.
The rigor and overall content quality of the trials described are good for the clinical focus of this review.
However, I believe the manuscript presents some issues that should be addressed prior to publication.
1. The title and abstract do not allow the readers to identify the focus of the review, which is summarizing, and linking together, the results from recent clinical trials. I believe in both title and abstract, or at least in the abstract, the specific aspects described for the various therapeutic strategies should be stated.
2. The manuscript requires some corrections regarding syntax, spelling, and grammar.
Examples:
- Line 9: “targeting molecular” was likely meant to be “targeting molecules”
- Line 11: “surface proteins such as for example” > “surface proteins, such as…” or “surface proteins, for example…”
- Line 14: “pretty fast”, “pretty” is a little too colloquial, would the authors consider “quickly” or other synonyms?
- Line 17: “have been recently be published” > “have been recently published”
- Line 18: “uncurbable” > “incurable”
3. Some acronym requires the addition of the definition as the broader public might not be accustomed to the name.
Example:
- Line 92: “VenR” > “Venetoclax and Rituximab (VenR)”
4. Some statements are not supported by proper referencing.
Examples:
- Sentences ending at Line 71, Line 74, 77, 78…
5. The citations of updates during conferences (e.g., American Society of Hematology Annual Meeting) are often missing the associated reference.
Example: Sentence between line 117 and 121
6. The tables cited within the manuscript (Table 1, Table 2, Table 3) are not present in the main file nor associated as supplementary.
Comments on the Quality of English LanguagePlease see above.
Author Response
Dears Editor and reviewers,
thank you for your letter on 04 Mar 2023 informing that our manuscript was found of interest, pending the resolution of some minor issues.
Our paper was modified according to your and the reviewer suggestions as followed reported.
REVIEWER #2
Q1. This review properly summarizes part of the actual knowledges derived from clinical trials testing novel combinations and/or novel agents.
Overall, this review allows to gain a proper overview of the various therapeutic strategies under investigation, the associated side effects, and the subsets of patients most likely taking advantage from the described therapies. Hence, it is likely to address a broad public.
The rigor and overall content quality of the trials described are good for the clinical focus of this review.
However, I believe the manuscript presents some issues that should be addressed prior to publication.
A1. We really appreciate the consideration of this reviewer.
Q2. The title and abstract do not allow the readers to identify the focus of the review, which is summarizing, and linking together, the results from recent clinical trials. I believe in both title and abstract, or at least in the abstract, the specific aspects described for the various therapeutic strategies should be stated.
A2. We would like to thank this reviewer and we modified the abstract according to his/her suggestions.
Q3. The manuscript requires some corrections regarding syntax, spelling, and grammar.
Examples: - Line 9: “targeting molecular” was likely meant to be “targeting molecules”
- Line 11: “surface proteins such as for example” > “surface proteins, such as…” or “surface proteins, for example…”
- Line 14: “pretty fast”, “pretty” is a little too colloquial, would the authors consider “quickly” or other synonyms?
- Line 17: “have been recently be published” > “have been recently published”
- Line 18: “uncurbable” > “incurable”
A3. Following the reviewer suggestion, we revised the manuscript carefully, corrected all the typos and flaws. We apologize for the mistakes.
Q4. Some acronym requires the addition of the definition as the broader public might not be accustomed to the name.
Example: - Line 92: “VenR” > “Venetoclax and Rituximab (VenR)”
A4. We apologize for this mistake. Definitions of all the acronyms were added.
Q5. Some statements are not supported by proper referencing.
Examples: - Sentences ending at Line 71, Line 74, 77, 78…
A5. Following the reviewer suggestion, we added references also in the legend to Figure 1.
Q6. The citations of updates during conferences (e.g., American Society of Hematology Annual Meeting) are often missing the associated reference.
Example: Sentence between line 117 and 121.
A6. Following the reviewer suggestion, we addended more references. Please see revised references #18, #20, #25, #28, #34, #53, #60, #61, #62, #63.
Q7. The tables cited within the manuscript (Table 1, Table 2, Table 3) are not present in the main file nor associated as supplementary.
A7. We apologize with the reviewer but likely there was an informatic issue, as in the submission there were 3 tables summarizing all the analyzed clinical trials. Please see tables 1-2-3.
Reviewer 3 Report
Comments and Suggestions for Authors
This review is interesting with clinical significance. Innovative combination, cellular therapy and bispecific antibodies had revolutionized CLL therapy. This is a comprehensive review. The followings are some comments to the authors.
1.I suggest the authors provide information on the epidemiology, clinical manifestations, the efficacy of standard treatment and prognosis of CLL in the section of INTRODUCTION.
2.Please define all abbreviations in the text when used for the first time. For example, line 21“MRD”.
3.Table 1,table 2,table 3 is not shown in the manuscript. Those are very important information, please provide the tables.
4. I recommend that authors provide safety information of clinical trials whenever possible, such as CAPTIVATE, VERITAS.
5.The discussion and conclusion can be improved. I suggest the authors add some information, for example, the prospects of CAR-Tand comparison of monoclonal antibody and bispecific antibodies.
Author Response
Dears Editor and reviewers,
thank you for your letter on 04 Mar 2023 informing that our manuscript was found of interest, pending the resolution of some minor issues.
Our paper was modified according to your and the reviewer suggestions as followed reported.
REVIEWER #3
Q1. I suggest the authors provide information on the epidemiology, clinical manifestations, the efficacy of standard treatment and prognosis of CLL in the section of INTRODUCTION.
A1. I agree with the reviewer however, this information will be reported by other colleagues in other invited reviews for the same special issue. This is why we decide to skit them.
Q2. Please define all abbreviations in the text when used for the first time. For example, line 21“MRD”.
A2. All abbreviations were defined, We apologize for the mistake.
Q3. Table 1,table 2,table 3 is not shown in the manuscript. Those are very important information, please provide the tables.
A3. We apologize with the reviewer but likely there was an informatic issue, as in the submission there were 3 tables summarizing all the analyzed clinical trials. Please see tables 1-2-3.
Q4. I recommend that authors provide safety information of clinical trials whenever possible, such as CAPTIVATE, VERITAS.
A4. Following the reviewer suggestion data on safety were added in the revised manuscript.
Q5. The discussion and conclusion can be improved. I suggest the authors add some information, for example, the prospects of CAR-T and comparison of monoclonal antibody and bispecific antibodies.
A5. Following the reviewer suggestion we revised the discussion addressing future perspectives, accessibility and sustainability of novel drugs.
Round 2
Reviewer 1 Report
Comments and Suggestions for Authors
The authors have appropriately addressed the specific concerns noted previously.
Comments on the Quality of English Language.
Author Response
Thank you so much.
Reviewer 2 Report
Comments and Suggestions for Authors
The authors have replied to each comment and made the proper major amendments to the manuscript.
I believe there are still some minor fixes in the context of orthography and spelling.
However, being stylish nuances, rather than content, I do not have any other major concern.
Examples of still existing or newly added spelling errors using the lining count as per the word file:
1) Page 1 Line 9: “moleculs” > ”molecules”
2) Page 1 Line 16-18: “Recently a large number of data….. have been recently published….” > “Recently a large number of data….. have been published….”
3) Page 1 line 21: “an measurable”
4) Page 5 line 199: “Jun” > “June”
5) Page 9 line 403-404: “BsAbs are similar to IgG and determines”
6) Page 10 line 444: “high‐risk R/R CLL and BsAbs syndrome” was meant to be Richter’s? In that case, the follow-up presentation at iwCLL is the one showing results on Richter’s transformation data, cited and described later.
7) Page 11 line 535: “standard or care” > “of”
Comments on the Quality of English LanguagePlease see above
Author Response
Q1 I believe there are still some minor fixes in the context of orthography and spelling. Examples of still existing or newly added spelling errors using the lining count as per the word file:
-Page 1 Line 9: “moleculs” > ”molecules”
-Page 1 Line 16-18: “Recently a large number of data….. have been recently published….” > “Recently a large number of data….. have been published….”
-Page 1 line 21: “an measurable”
-Page 5 line 199: “Jun” > “June”
-Page 9 line 403-404: “BsAbs are similar to IgG and determines”
-Page 10 line 444: “high‐risk R/R CLL and BsAbs syndrome” was meant to be Richter’s? In that case, the follow-up presentation at iwCLL is the one showing results on Richter’s transformation data, cited and described later.
-Page 11 line 535: “standard or care” > “of”
A2. We kindly thank the work done by this reviewer who carefully reviewed the manuscript. Typos were corrected.
Reviewer 3 Report
Comments and Suggestions for Authors
Some comments for table 1:
1. What does "yy" in the PFS column mean? Please explain.
2. In CAPTIVATE and FLAIR, which group do the two lines of efficacy data belong to? In CLL13,GLOW and CLL14, Which group does the efficacy data belong to? Plaese confirm that. I suggest standardizing the presentation format of the data of randomized controlled trial in Table 1.
3.I suggest adjusting the size of the table to try to avoid splitting words onto separate lines.
Author Response
Q1. What does "yy" in the PFS column mean? Please explain.
A1 “yy” means years and “y” year. They were added below the tables.
Q2. In CAPTIVATE and FLAIR, which group do the two lines of efficacy data belong to? In CLL13,GLOW and CLL14, Which group does the efficacy data belong to? Plaese confirm that. I suggest standardizing the presentation format of the data of randomized controlled trial in Table 1.
Q3. I suggest adjusting the size of the table to try to avoid splitting words onto separate lines.
A2-3. We apologize for these misunderstandings. We reformatted Table 1 according to the reviewer suggestions and we agree that it is clearer.
Round 3
Reviewer 3 Report
Comments and Suggestions for Authors
This manuscript can be accepted in present form